# When optimization for governing human-environment tipping elements is neither sustainable nor safe

Wolfram Barfuss[1,2], Jonathan F. Donges[1,3], Steven J. Lade[3,4] & Jürgen Kurths[1,2,5]

Optimizing economic welfare in environmental governance has been criticized for delivering short-term gains at the expense of long-term environmental degradation. Different from economic optimization, the concepts of sustainability and the more recent safe operating space have been used to derive policies in environmental governance. However, a formal comparison between these three policy paradigms is still missing, leaving policy makers uncertain which paradigm to apply. Here, we develop a better understanding of their interrelationships, using a stylized model of human-environment tipping elements. We find that no paradigm guarantees fulfilling requirements imposed by another paradigm and derive simple heuristics for the conditions under which these trade-offs occur. We show that the absence of such a master paradigm is of special relevance for governing real-world tipping systems such as climate, fisheries, and farming, which may reside in a parameter regime where economic optimization is neither sustainable nor safe.

[1] Potsdam Institute for Climate Impact Research, 14473 Potsdam, Germany. [2] Department of Physics, Humboldt University, 12489 Berlin, Germany. [3] Stockholm Resilience Centre, Stockholm University, 11419 Stockholm, Sweden. [4] Fenner School of Environment and Society, The Australian National University, Canberra, ACT 2601, Australia. [5] Saratov State University, Saratov 410012, Russia. Correspondence and requests for materials should be addressed to W.B. (email: barfuss@pik-potsdam.de)

The Sustainable Development Goals[1] and the Paris climate agreement set the target of prosperous development for people and our planet. Yet, it remains challenging to translate these aims into concrete policy implementations, accounting for non-linearities, such as tipping elements[2,3], regime shifts[4,5], and multi-stabilities[6], as well as multiple kinds of uncertainties[7–9], and extreme events[10].

To support the decision making processes in these contexts, we ask the question how the three prominent decision making paradigms of economic welfare optimization, sustainability and safe operating space compare with each other. Specifically, we investigate the parameter regimes for synergies and trade-offs when applying these paradigms to the management of tipping elements[11] and how these findings relate to the three real-world systems of climate, fisheries and farming.

Optimization approaches have emerged as the primary guiding principle to derive a policy strategy for environmental governance[12,13]. Most often, the present value of macroeconomic social welfare, i.e., the sum of discounted future benefits minus costs, is the target to be optimized. Such optimization approaches have been criticized regarding the discount rates used, delivering short term gains at the expense of long-term environmental degradation[14,15]. Further criticism targets the lack of a systems perspective required to understand the structural landscape of model dynamics, as well as the assumptions made due to imperfect information[6,9,10]. This critique is partly dealt with in optimization variants, such as robust[7,16] or viable[17–19] control, which are dealing with multiple types of uncertainty[20]. Naturally, other or multiple objectives[21] and criteria[22,23] with possible constraints[24] can be optimized as well. In this work, we use the term solely in the narrow economic sense of maximizing the present value as defined in Eq. 1 below.

In recognition of increasing environmental and social threats[25] the policy paradigm of sustainability has emerged in the scientific and political discourse[26,27]. The economics of sustainability has brought up many definitions of sustainability alone[28–31]. In these analyses sustainability is usually imposed as a constraint within an economic welfare optimization paradigm. Trade-offs to economic welfare optimization are well known[28,32]. However, these classic social welfare optimization approaches are challenged through the increasing recognition of non-linearities, such as tipping points, regime shifts, uncertainties and the risk of catastrophic outcomes[6,9]. Taking up these challenges, e.g., non-convexities[33] and climate tipping elements[34,35] have been studied within an economic framework. Here, we derive our formal definition of sustainability from the Brundtland report[26]. Its design is deliberately simple and targeted to the mathematical framework we use (see below). We do not intend our definition to be applicable to a general model of a welfare economy[12,27].

Recent advances in sustainability science have brought forth tolerable windows[36] or safe operating spaces[37,38] as a policy paradigm to derive concrete actions from[39]. These concepts originate from resilience thinking[40] and a precautionary principle[41] to deal with potential dangerous tipping elements in the environmental governance system. Trade-offs but also synergies with optimization thinking have been therefore discussed[42]. Also formal analyses studying relations between resilience as a system property and sustainability were conducted[43,44].

However, the reciprocal relationships between these three paradigms of economic optimization, sustainability and safe operating space is still insufficiently explored. Such an understanding is important in order to judge, for example, when economic optimization is, or is not, an appropriate policy goal. Also, guidance is required when a sustainability paradigm may conflict with a safe operating space paradigm and vice versa.

Here, we report progress towards a better understanding of the mutual relationships between these three paradigms of economic optimization, sustainability and safe operating space by applying them to a stylized model of a human-environment tipping element. We do so because of the increasing importance of tipping points and regime shifts in environmental governance. Our model is deliberately stylized, thereby applicable across multiple cases and scales, to gain a deeper understanding more complex models might miss. The formal definitions of the three paradigms are designed to fit our mathematical framework (see below). Since we do not focus on intragenerational justice in this article, one agent suffices as a decision making subject, in contrast to a multiagent setting. We find that there exists no master paradigm between the three examined, i.e., a policy can be any combination of optimal or not, sustainable or not and safe or not. This is of special relevance to the climate system which may reside at the edge in the parameter regime where economic welfare optimization becomes neither sustainable nor safe. This suggests the use of more advanced paradigms to support decision making in climate policy.

## Results

**Stylized model of a human-environment tipping element.** We use the mathematical framework of Markov Decision Processes[45,46], in which an agent makes decisions about how to interact with its environment (Fig. 1a). Our particular environment can reside in either a prosperous state, which provides immediate rewards (also called payoffs) to the agent, or a degraded state, from which the agent receives no payoff. At each time step, the agent chooses between two actions $a$, exerting either a high or low pressure on the environment. Depending on the current state $s$, the current action $a$ and the subsequent state $s'$, the agent receives an immediate reward $r$ (Fig. 1b). At the prosperous state, taking the low pressure action the agent is guaranteed to receive reward $r_l$ and remain at the prosperous state. However, taking the high pressure action, the agent may receive reward $r_h$ (which is typically larger than $r_l$), but risks triggering a collapse of the environment to the degraded system state with non-zero probability $\delta$ and no immediate reward at all. From there, only the low pressure action opens the option to recover to the prosperous state with non-zero probability $\rho$.

For example, the high pressure action could correspond to emitting a business-as-usual amount of carbon to the atmosphere yielding a reward of high, short-term economic output as long as the system has not tipped. The low pressure action resembles emitting a reduced amount of carbon, assuming a lower short-term economic output for the guarantee to not trigger climate tipping elements into a disastrous state.

A policy $\pi$ is a function that specifies what action $a$ to apply at a system state $s$. The agent receives reward $r_t$ at time step $t$. The value $v_\pi(s)$ of a state $s$ under a given policy $\pi$ is given by the expected value of the normalized accumulated discounted rewards $r$ with discount factor $0 \le \gamma \le 1$ when starting in state $S_0 = s$ and following policy $\pi$:

$$v_\pi(s) = \mathbb{E}_\pi \left[ \lim_{T \to \infty} \frac{\sum_{t=0}^{T} \gamma^t r_t}{\sum_{t=0}^{T} \gamma^t} | S_0 = s \right]. \quad (1)$$

Note that the discount factor actually denotes the farsightedness of the agent. Thus, $\gamma = 1$ corresponds to no discounting (weighting all rewards equally regardless of when they are

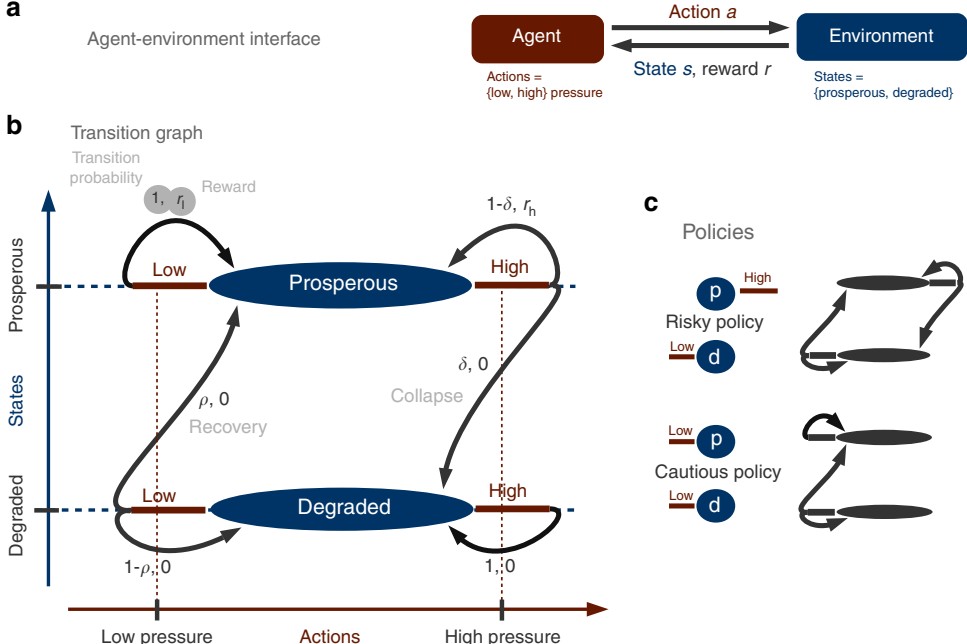

**Fig. 1** Conceptual model of a human-environment tipping element. **a** Agent-environment interface: based on the state information and received reward, the agent chooses an action $a$ from its actions set to gain rewards. **b** The transition graph gives state transition probabilities and corresponding rewards for all triples of state $s$, action $a$, next state $s'$, i.e., in state $s$ the agent takes action $a$ and moves to state $s'$. **c** Risky and cautious policies including the resulting Markov chains as a transition graph

expected), whereas $\gamma = 0$ corresponds to completely myopic, fully discounting agents.

**Paradigm definitions**. We classify policies according to whether they are economic welfare optimal or not, sustainable or not, and safe or not. For the sake of simplicity we focus on two deterministic policies, distinguishing whether the agent should apply the low or the high pressure action at the prosperous state (Fig. 1c): the risky policy ($\pi_r(p) = h$, $\pi_r(d) = l$), applying the high pressure action at the prosperous state and the low pressure one at the degraded state and the cautious policy ($\pi_c(p) = l$, $\pi_c(d) = l$), applying the low pressure action at the prosperous, as well as the degraded state.

A policy $\pi$ is defined as optimal (in the economic welfare sense) if its value $v_\pi(s)$ (Eq. 1) for every state s is larger than or equal to the value of any other policy[46].

Based on the Brundtland Commission's report on sustainable development[26] a sustainable policy should fulfill two requirements: First, meet the needs of the present. We translate this formally into the agent evaluating the present state $s$ as acceptable (similar to viable[17], tolerable[36] or desirable[47]), if its value (Eq. 1) exceeds a normatively chosen minimum acceptable value $r_{min}$:

$$s \text{ acceptable under } \pi \text{ iff } v_\pi(s) \geq r_{min} \qquad (2)$$

Note, that the division of state space into acceptable and unacceptable states is not identical for all polices, but depends on the rewards receivable through executing a policy. Second, a sustainable policy should sustain the ability to meet the needs of the future[26].

We define a policy $\pi$ as sustainable if every state the agent eventually visits under policy $\pi$ is acceptable (Eq. 2).

Note that this reduction of sustainability to the one-dimensional value $v_\pi(s)$ has much similarity with the notion of weak sustainability[48].

The Safe Operating Space (SOS)[37] is typically defined as a subset of the whole state space $\mathcal{S}$, containing favorable system states bounded by thresholds[39,49]. In practice, the position of these potential tipping thresholds is always uncertain and the boundaries are placed at the lower end of the uncertainty zone. In that way the definition of the safe operating space states constitutes a normative judgment about the risk the decision maker is willing to tolerate. In the subsequent analyses we take the extreme position of no risk tolerance and identify the SOS with only the (more favorable) prosperous state, independent of the collapse probability $\delta$.

We define a policy $\pi$ as safe if every state the agents eventually visits under policy $\pi$ lies within the SOS.

In contrast to acceptable and unacceptable states, safe states are independent of the policy used.

In summary, our stylized model of a human-environment tipping element depends on the five parameters $\delta$, $\rho$, $\gamma$, $r_l/r_h$, $r_{min}/r_h$: the probability of a collapse from the prosperous to the degraded state under the high pressure action $\delta$, the probability of recovery from the degraded to the prosperous state under the low pressure action $\rho$, the agent's discount factor $\gamma$, the high reward receivable from the high pressure action when staying at the prosperous state $r_h$, the low reward receivable by taking the low pressure action at the prosperous state $r_l$, and the normatively chosen minimum acceptable reward $r_{min}$ a state value must have to be perceived as acceptable under a certain policy. Since all three rewards come in arbitrary units, the policy classification only depends on their ratios.

**Classification of risky and safe policy**. Based on Eqs. 1 and 2 we analytically compute whether the risky and the cautious policy are

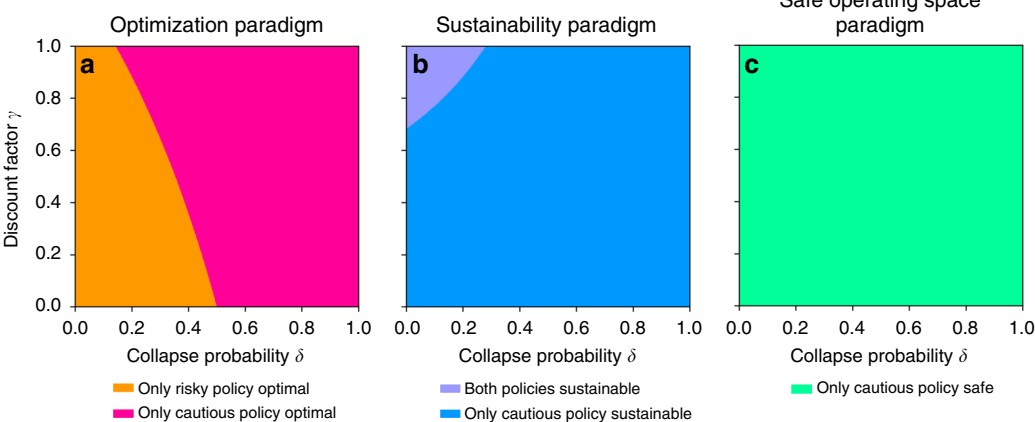

**Fig. 2** Classification of the risky and cautious policy according to the three policy paradigms: **a** optimization, **b** sustainability, and **c** safe operating space in the model parameter space (shown here as collapse probability $\delta$ vs. discount factor $\gamma$); remaining parameters were chosen as $\rho = 0.2$, $r_l/r_h = 0.5$, $r_{min}/r_h = 0.3$ for illustration purposes. Colored regions result from analytically derived equations (see Methods). Depending on the parameter region, both risky and cautious policy can be optimal and sustainable. Only the cautious policy is safe

optimal or not, sustainable or not and safe or not depending on the model parameters ($\delta$, $\rho$, $\gamma$, $r_l/r_h$, $r_{min}/r_h$) (see Methods and Fig. 2).

We observe that above a certain critical value of the collapse probability $\delta$ the cautious policy becomes optimal (Fig. 2a, pink), despite the smaller immediate reward $r_l = 0.5r_h$. This result confirms previous findings on optimal management with regime shifts[50].

Further, we find a decreasing critical collapse probability with increasing farsightedness $\gamma$. Hence, for more farsighted societies the risky policy is optimal only for small collapse probabilities $\delta$ (orange).

Provided the low pressure reward exceeds the normative minimum acceptable value threshold, $r_l \geq r_{min}$, then the cautious policy is sustainable for all parameter combinations $\delta$, $\rho$, $\gamma$, $r_l/r_h$ (Fig. 2b, blue and purple). Only for small collapse probabilities $\delta$ and simultaneously high farsightedness $\gamma$ the risky policy becomes sustainable as well (purple). This is because in this parameter region the risky policy is acceptable also at the degraded state (Methods).

The cautious policy is a safe policy independently from the parameter combinations $\delta$, $\rho$, $\gamma$, $r_l/r_h$, $r_{min}/r_h$ (Fig. 2c, green). It is important to emphasize that there is no combination of parameters at which the risky policy is safe.

**Relationships between paradigms.** We find that policies can be classified along all logical combinations of the three examined paradigms (optimization, sustainability, safe operating space). This yields a classification of policies into eight different categories (Fig. 3).

In particular, optimal policies are not necessarily sustainable (opt and not sus: Fig. 3, red and yellow). This is the case if the normative value threshold $r_{min}$ is too large. The cautious policy does not return enough value to be sustainable ($r_l < r_{min}$, yellow) and the risky policy at the degraded state produces too little future reward to be sustainable, due to the low chance of recovery and lack of farsightedness.

Nor are optimal policies necessarily safe (opt and not safe: Fig. 3, red and purple). This occurs in parameter regions where the risky policy is optimal. The risky policy cannot be safe because of the risk of collapse to the degraded state.

A safe policy does not necessarily imply a sustainable policy either (safe and not sus: Fig. 3, green and yellow). When the

normative threshold value for sustainability $r_{min}$ exceeds the reward from a low pressure action $r_l$: $r_{min} > r_l$, then the cautious policy is safe but not sustainable. Following a similar line of argument, the SOS concept[37] has been extended to a Safe And Just Operating Space (SAJOS) which additionally accounts for social indicators[51], such as the number of people living in extreme poverty. Thus, SAJOS policies can be interpreted as the overlap of safe with sustainable policies. Within our model, we can give a definite criterion for when this form of SAJOS exists: as long as the reward from a low pressure action $r_l$ exceeds the normative threshold value $r_{min}$ ($r_l > r_{min}$), the cautious policy is both safe and sustainable (Fig. 3, cyan and gray).

However, there exist also sustainable policies outside the SOS (sus and not safe: Fig. 3, blue and purple.) These are risky policies (hence, not safe) with simultaneously high farsightedness $\gamma$ and low collapse probability $\delta$. At those parameter regions the degraded state is still evaluated as acceptable due to sufficient anticipated future rewards and therefore the risky policy is sustainable. The circumstance that parameter regimes exist that are sustainable but not safe and vice versa clearly stems from our definition of sustainability which resembles a form of weak sustainability[48]. By doing so we can conceptually separate issues of environmentally safe and socially just without compromising the target of a safe and just parameter space regime.

Note that this classification into the eight different policy paradigm combinations also applies to the case of absolute farsightedness ($\gamma = 1$; see the tops of Fig. 3b–e). Thus, the trade-offs between the examined paradigms do not vanish, as one might presume considering the debate about appropriate discount rates[14,52].

**Volume of paradigm combinations.** So far, we have visualized the parameter space of our stylized tipping element model in two dimensional sections and fixed the remaining parameters for illustrative purposes. By doing so, we showed the mutual dependence between parameters, foremost the discount factor $\gamma$ and the collapse probability $\delta$. However, in the light of considerable parameter uncertainty we ask how large the eight regimes of paradigm combinations are, given the whole parameter space (Fig. 4).

We observe the most likely option to be the regime that is neither optimal, neither sustainable nor safe followed by the parameter sweet spot regime in which all paradigms yield the

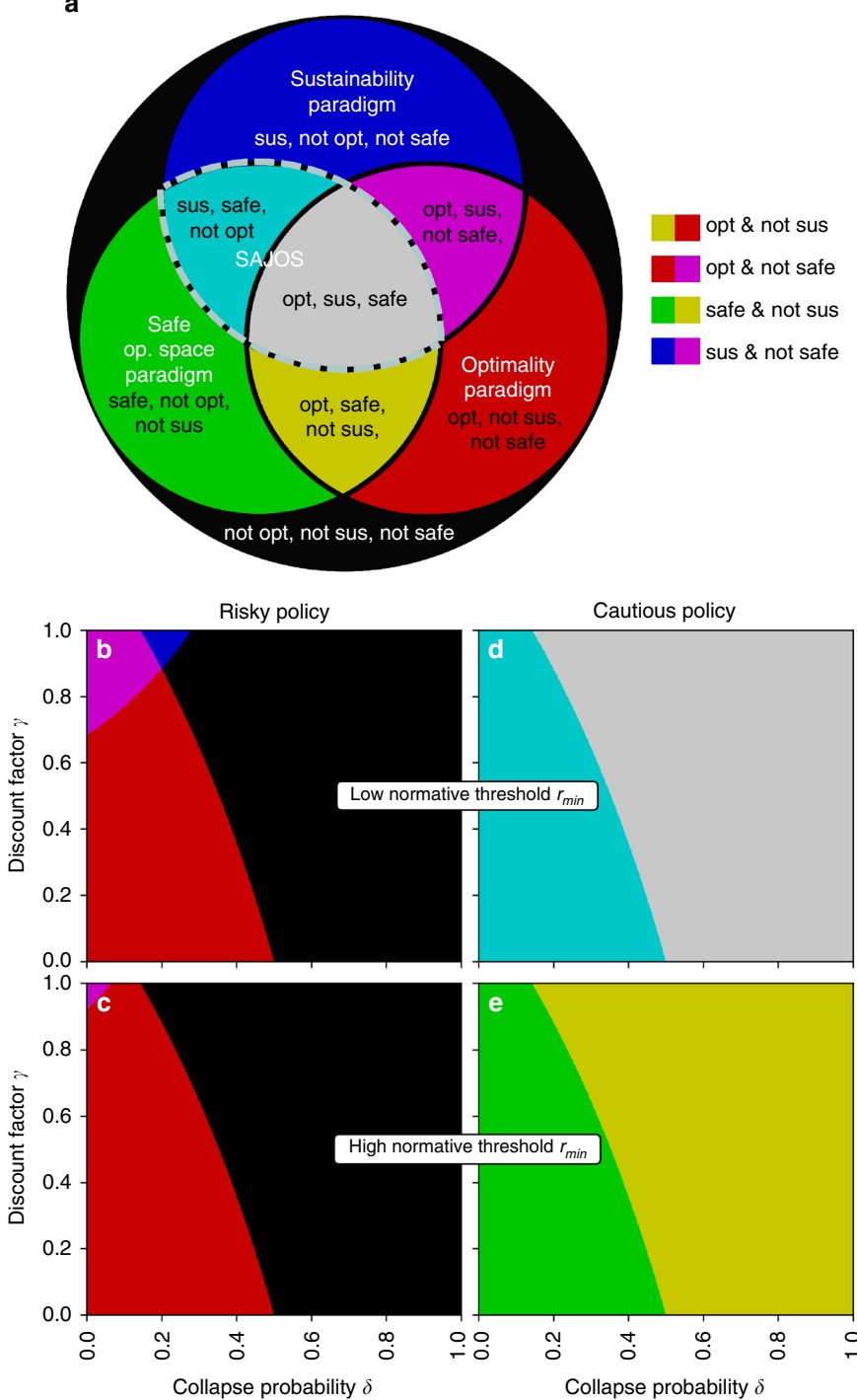

**Fig. 3** Paradigms classification for risky and cautious policy. There exist policies in parameter space of our model for all logical combinations of paradigm classifications. **a** i.e. a policy can be any combination of (not) optimal, (not) sustainable and (not) safe. Remaining parameters where chosen as $\rho = 0.2$, $r_l/r_h = 0.5$ for illustration purposes (c.f. Methods). For a sufficiently low normative threshold value $r_{min} \leq r_l$ (here $r_{min}/r_h = 0.3$) a Safe And Just Operating Space (SAJOS) exists, which we identified as the overlap of safe and sustainable policies **b**, **d** (gray and cyan area). For a sufficiently large $r_{min} > r_l$ (here $r_{min}/r_h = 0.7$) a SAJOS does not exist **c**, **e**

cautious policy as optimal, sustainable and safe. Together they constitute a parameter space volume of approx. 45% in which the three paradigms of economic optimization, sustainability and safe operating space align with each other in yielding the same policy. Interestingly, the third likeliest option is the paradigm combination in which the risky policy is optimal but neither sustainable nor safe. This is the most likeliest parameter regime among those

where the paradigms yield different policies. Thus, blindly applying economic optimization in a our stylized tipping element has a significant chance of leading to policies that are neither sustainable nor safe.

On the other hand, the volume of the safe and just operating space (gray and cyan bars in Fig. 4) is comparable to the most likeliest (black) regime. Thus, about one out of four random

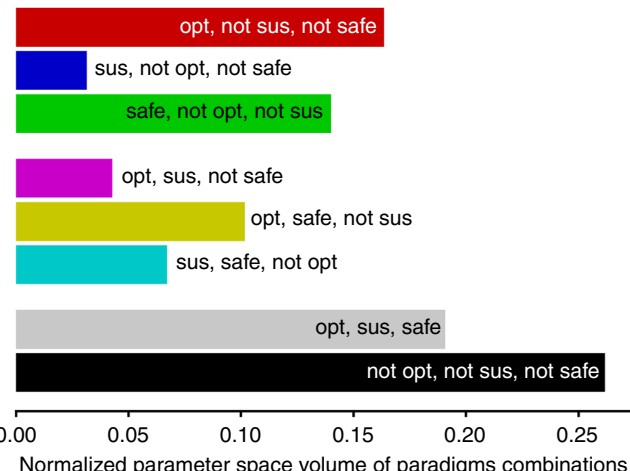

**Fig. 4** Ratios of parameter space volumes for all eight paradigms combination. All parameters ($\delta$, $\rho$, $\gamma$, $r_l/r_h$, $r_{min}/r_h$) were chosen linearly between 0 and 1 for both the risky and the cautious policy. As a direct consequence of our definitions of the safe operating space paradigm and the cautious and risky policy, all paradigm combinations which are safe correspond to the use of the cautious policy, in all others the risky policy was applied. A random decision making agent within a random tipping element will most likely end up with a policy that is neither optimal, neither sustainable nor safe, followed by the parameter sweet spot regime where the policy is simultaneously optimal, sustainable and safe. Interestingly, the third likeliest option is a parameter regime which is optimal, but neither sustainable nor safe

decision making agents interacting with a random tipping element will end up in the safe and just operating space.

**Application to real-world human-environment tipping elements**. The above policy classification offers valuable insights for the governance of real-world human-environment systems. We discuss how our analysis relates to the cases of the climate system, fisheries and farming. Our purpose is to gain a qualitative understanding how our model relates to important real-world challenges of environmental governance, not a detailed assessment of the latter. Therefore, we roughly estimate the respective collapse and recovery probabilities per time step $\delta$ and $\rho$ of our model via the typical timescales on which these systems remain in one state or the other (see Methods). Additionally, we added a parameter sensitivity analysis by visualizing the likelihood of ending up in a certain parameter regime by color gradients between regimes (Fig. 5).

Regarding the climate system, we acknowledge that several interacting tipping elements contribute to the system's behavior[2] and its representation as a single tipping element is a huge simplification on its own. Nevertheless, we assume that the current state of the climate system is still comparable to the prosperous one of our model and relevant timescales for triggering a collapse of 30 to 50 years under business-as-usual socio-economic development scenarios[2,53,54]. Regarding the recovery timescale it has been shown that human perturbations of the climate system already changed its trajectory on a multi-millennial timescale[55,56]. Therefore we assume a recovery probability per time step $\rho$ close to zero (Fig. 5).

For sufficiently large collapse probabilities (collapse time scale near 20 years and smaller), the climate system is likely to reside in a parameter sweet spot (gray area), where applying an optimization, sustainability or SOS paradigm results in the cautious policy as the advisable way of governing the climate system. However, if the collapse probability per time step is smaller (collapse time

scale near 50 years and larger) the situation is different. Here, an SOS and a sustainable paradigm would still yield the cautious policy (Fig. 5, cyan), but an optimization paradigm is likely to give the risky policy (Fig. 5, red), which at this point is neither sustainable nor safe. We conclude that in climate policy, economic welfare optimization alone may neither be sustainable nor safe.

For fishery systems, both transition probabilities certainly depend on a variety of factors, e.g., fisher's technical and cultural traits or the dominant fish species in the system, as well as external factors such as climate change influencing habitat condition[57,58]. The timescale of a fisheries collapse has been shown to lie within decades[59]. Roughly consistent with observational and modeled data from the Baltic sea, where the stable regime of high cod biomass lasted approximately from 1970 to 1990[57,60], we assume a typical collapse timescale of around 20 years. Concerning the typical recovery time scale, successful attempts of fish stocks recovery lasted for decades[61], but is estimated to generally exceed this duration[62]. We therefore assume a larger typical recovery timescale of around 50 years. The color gradient in Fig. 5 at the fisheries point does not clearly single out a paradigms regime, indicating the dependence on the other parameters at this point. A risky policy might be economically optimal (Fig. 5, red), but leads eventually to the collapse of fish stock (c.f[59].). At the collapsed and degraded state the conditions for the fishers are not acceptable. Therefore they have to leave the system and cannot wait for the fish's recovery. But further investigation is needed to reduce the uncertainty with respect to the other parameters.

Last, we look at the case of land degradation by farming in our stylized model. Land degradation and restoration is a complex topic with many influencing factors[63]. Nevertheless, land degradation by farming has been identified as a tipping element by Kinzig and others[64], where the authors discuss the case of the western Australian wheatbelt with a typical collapse timescale of about 100 years. Soil recovery is estimated to take place within 20 to 1000 years[65], which is roughly consistent to Kitzing et al., where the duration to reach equilibrium again is estimated with up to 300 years. We therefore assume a typical recovery timescale of about 300 years. In contrast to climate and fisheries, the transition probabilities we associated with the process of land degradation by farming suggest, that here an optimality paradigm is very likely to yield the risky policy which is neither sustainable nor safe despite considerate parameter uncertainty (red area in Fig. 5).

Taken together, it is interesting to see that in particular the climate system may reside at the edge of the parameter regime where economic welfare optimization becomes neither sustainable nor safe (Fig. 3). For land degradation by farming, our assessment suggests that an optimal policy is likely to yield a non-sustainable and non-safe policy whereas for fisheries the situation is less clear.

## Discussion

Overall, our results show that there exists no master paradigm among the three examined in our model of environmental governance of a stylized tipping element. Policies can be classified by any combination of optimal, sustainable and safe. A master paradigm, in contrast, would guarantee fulfilling requirements imposed by other paradigms. Consequently, the selection of appropriate policy paradigms, especially in more complex settings and models, can be critical for effective environmental governance.

Specifically, our results show theoretically, as well as empirically that economic welfare optimization for managing tipping

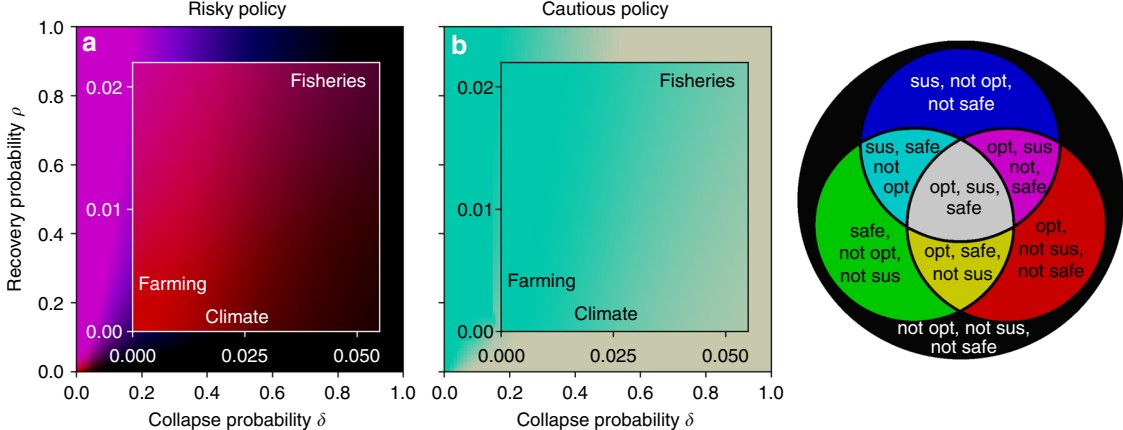

**Fig. 5** Human-environment systems in paradigms classification. For risky (**a**) and (**b**) cautious policy here shown in model parameter space of collapse probability $\delta$ versus recovery probability $\rho$. Color indicates the paradigms combination similarly as in Fig. 3. Here, additional gradual changes between the color regimes indicate the probability of being in a certain paradigms combinations regime under parameter uncertainty ranges. Remaining parameters where chosen linearly within the range of $0.95 \leq \gamma \leq 0.99$, $0.3 \leq r_l/r_h \leq 0.7$, $0.1 \leq r_{min}/r_h \leq 0.5$. The approx. transition probabilities $\delta$ and $\rho$ were assigned to the human-environment systems climate, fisheries and farming agriculture according to the timescale of the average time spent in one state (see Methods). For farming, a risky policy is likely to be optimal but neither sustainable nor safe. The parameter uncertainty of the other parameters does not allow a clear statement in which parameter regime fisheries are likely to fall. The climate system may lie at the edge of the sweet spot, where all paradigms yield the cautious policy. However, for smaller collapse probability $\delta$ optimization is more likely to yield the risky policy, which becomes also neither sustainable nor safe at this point. This suggests the use of other paradigms for climate policy making

elements may be neither sustainable nor safe. For example, the volume of the corresponding paradigm combination in parameter space is the largest among those in which the three paradigms actually yield different policies. This suggests the conclusion that the mere structure of a tipping element causes a comparable high chance of obtaining a policy that is neither sustainable nor safe when blindly following an optimization paradigm. On the other hand, our model also indicates parameter regimes where economic optimization can safely and sustainably be used.

We derived simple heuristics to anticipate when a policy is economic welfare optimal, sustainable and safe. A risky policy may be optimal when the probability of collapse and/or the far-sightedness are sufficiently small. It may be sustainable when the probability of a collapse is sufficiently small but the farsightedness is sufficiently large. However, it cannot be safe. A cautious policy may be optimal when the collapse probability and/or the far-sightedness are sufficiently large. It is sustainable if its immediate reward exceeds the normatively chosen minimum acceptable reward and it is always safe. The absence of a master paradigm is of special relevance for governing the climate system, since the latter may reside at the edge between parameter regimes where economic welfare optimization becomes neither sustainable nor safe.

Extensions are possible in many directions. Constrained optimization[24] is a straight-forward way to combine the paradigms examined. Policy makers could aim for the maximum economic welfare delivering a policy that is safe and sustainable, or least-cost safe target strategies[15]. This is certainly a better approach than relying on economic welfare optimization alone for model-based policy advice. Examples of models for policy advice certainly include integrated assessment models or the use of the maximum sustainable yield in fisheries management. However, one might not desire to obtain the welfare optimal safe and sustainable policy but e.g., the most resilient one, which calls for an operationalization of modern social-ecological resilience concepts[66].

The application of our model to real-world systems in this article is of qualitative, illustrative nature. A more detailed

analysis of real world tipping elements in which typical transition probabilities might be estimated from empirical time series could be a way forward to systematize and draw lessons from the multitude of human-environmental tipping elements[67].

Applying our analyses to larger, more complex Markov decision processes would be a way to extend the understanding of the relationships between the paradigms examined. Moreover, it may be desirable to include further policy paradigms into the analyses, e.g., aiming for a large option space of future decision makers[30,68]. Based on such analyses, policy makers could make better informed decisions on how to translate the Sustainable Development Goals and the Paris climate agreement into concrete policy implementations.

## Methods

**Derivation of value functions**. There are four deterministic policies in our Markov decision process model: (1) $\pi_r(p) = h$, $\pi_r(d) = l$, (2) $\pi_c(p) = l$, $\pi_c(d) = l$, (3) $\pi_3(p) = h$, $\pi_3(d) = h$, (4) $\pi_4(p) = l$, $\pi_4(d) = h$. We concentrate on deterministic policies only to simplify the calculation without loss of generality, because if an optimal policy exits there also exists a deterministic optimal policy[46]. We further focus here only on the first two policies, named the risky and the cautious policy, since the remaining two apply a high pressure action at the degraded state. This will trap the agent at this position for eternity without receiving any reward. The math on these policies is left to the interested reader.

In the following we derive the analytical expressions of the state values of these policies as functions of the parameters ($\delta$, $\rho$, $\gamma$, $r_l$, $r_h$). From Eq. 1 and for $\gamma < 1$ one can derive the recursive relationship between state values, known as the Bellman Equation[69]:

$$v_\pi(s) = \sum_{s'} p(s'|s, \pi(s))[(1-\gamma)r(s, \pi(s), s') + \gamma v_\pi(s')] \quad (3)$$

with $p(s'|s, \pi(s))$ being the probability to enter state $s'$ given the agent has started in state s and applied action $\pi(s)$.

Applied to our model the value for the prosperous state reads

$$v_\pi(p) = \begin{cases} \delta\gamma v_\pi(d) + (1-\delta)[(1-\gamma)r_h + \gamma v_\pi(p)] & \text{for } a = h \\ (1-\gamma)r_l + \gamma v_\pi(p) & \text{for } a = l \end{cases} . \quad (4)$$

The value for the degraded state is given by

$$v_\pi(d) = \begin{cases} \gamma v_\pi(d) & \text{for } a = h \\ (1-\rho)\gamma v_\pi(d) + \rho\gamma v_\pi(p) & \text{for } a = l \end{cases}. \tag{5}$$

To obtain the explicit state values for the risky policy ($\pi_r(p) = h$, $\pi_r(d) = l$) we solve the system of equations

$$v_{\pi_r}(p) = \delta\gamma v_{\pi_r}(d) + (1-\delta)\big[(1-\gamma)r_h + \gamma v_{\pi_r}(p)\big] \tag{6}$$

$$v_{\pi_r}(d) = (1-\rho)\gamma v_{\pi_r}(d) + \rho\gamma v_{\pi_r}(p), \tag{7}$$

which yields

$$v_{\pi_r}(p) = r_h \frac{(1-\delta)(1-(1-\rho)\gamma)}{1-(1-\delta-\rho)\gamma} \tag{8}$$

$$v_{\pi_r}(d) = r_h \frac{(1-\delta)\rho\gamma}{1-(1-\delta-\rho)\gamma}. \tag{9}$$

To obtain the explicit state values for the cautious policy ($\pi_c(p) = l$, $\pi_c(d) = l$) we solve the system of equations

$$v_{\pi_c}(p) = (1-\gamma)r_l + \gamma v_{\pi_c}(p) \tag{10}$$

$$v_{\pi_c}(d) = (1-\rho)\gamma v_{\pi_c}(d) + \rho\gamma v_{\pi_c}(p), \tag{11}$$

which yields

$$v_{\pi_c}(p) = r_l \tag{12}$$

$$v_{\pi_c}(d) = \frac{\rho\gamma r_l}{1-(1-\rho)\gamma}. \tag{13}$$

For $\gamma = 1$ we compute the values $v_\pi$ (which are independent from the initial state for $\gamma = 1$) by multiplying the stationary state of the effective Markov chain with the reward vector $\mathbf{r}^\pi \in \mathbb{R}^{|S|}$ whose components read

$$r_s^\pi = \sum_{s'} p(s'|s, \pi(s))r(s, \pi(s), s'). \tag{14}$$

The components of the transition matrix $\mathbf{P}^\pi$ of the effective Markov chain read

$$P_{s's}^\pi = p(s'|\pi(s), s). \tag{15}$$

The stationary state $\boldsymbol{\sigma}_\pi$ is the normalized eigenvector of the transition matrix with eigenvalue one. Hence,

$$v_\pi = \sigma_\pi \cdot \mathbf{r}^\pi. \tag{16}$$

Performing this calculation for risky and cautious policy explicitly yields consistent results with the calculation for $0 \le \gamma < 1$ from above. For $\gamma = 1$ the value $v_\pi$ can be obtained by simply inserting $\gamma = 1$ into Eqs. 8 and 9 for the risky policy and Eqs. 12 and 13 for the cautious policy.

**Analytical expressions for paradigm policy classification**. To derive the analytical expression of the hypersurface in parameter space that separates the regions where either the risky or the cautious policy is optimal we set $v_{\pi_r}(p) \overset{\text{set}}{=} v_{\pi_c}(p)$ (or equivalently $v_{\pi_r}(d) \overset{\text{set}}{=} v_{\pi_c}(d)$, since the parameter combination where a policy is optimal is independent from the state) and implicitly obtain

$$\tilde{r}_h \cdot \big(1-\tilde{\delta}\big)\big(1-\tilde{\gamma}(1-\tilde{\rho})\big) = \tilde{r}_l \cdot \Big(1-\tilde{\gamma}\big(1-\tilde{\delta}-\tilde{\rho}\big)\Big). \tag{17}$$

To obtain the hypersurface that separates state $s$ being acceptable from being not acceptable under policy $\pi$ we apply the definition from Eq. 2: $v_\pi(s) \overset{\text{set}}{=} r_{\min}$. Hence, for the risky policy at the prosperous state we set $v_{\pi_r}(p) \overset{\text{set}}{=} r_{\min}$ and obtain implicitly

$$\tilde{r}_h \cdot \big(1-\tilde{\delta}\big)\big(1-\tilde{\gamma}(1-\tilde{\rho})\big) = \tilde{r}_{\min} \cdot \Big(1-\tilde{\gamma}\big(1-\tilde{\delta}-\tilde{\rho}\big)\Big). \tag{18}$$

For the risky policy at the degraded state we set $v_{\pi_r}(d) \overset{\text{set}}{=} r_{\min}$ and obtain implicitly

$$\tilde{r}_h \cdot \big(1-\tilde{\delta}\big)\tilde{\rho}\tilde{\gamma} = \tilde{r}_{\min} \cdot \Big(1-\tilde{\gamma}\big(1-\tilde{\delta}-\tilde{\rho}\big)\Big). \tag{19}$$

For the cautious policy at the prosperous state we set $v_{\pi_c}(p) \overset{\text{set}}{=} r_{\min}$ and obtain implicitly

$$\tilde{r}_l = \tilde{r}_{\min}. \tag{20}$$

For the cautious policy at the degraded state we set $v_{\pi_c}(d) \overset{\text{set}}{=} r_{\min}$ and obtain implicitly

$$\tilde{r}_l \cdot \tilde{\rho}\tilde{\gamma} = \tilde{r}_{\min} \cdot (1-\tilde{\gamma}(1-\tilde{\rho})) \tag{21}$$

To get from acceptability to sustainability for the risky policy one has to logically combine Eqs. 18 and 19. The risky policy is sustainable only if both the prosperous and the degraded state are acceptable since it will visit both states recurrently. The safe policy is sustainable exactly where the prosperous state is acceptable since it will eventually end up and remain at the prosperous state. Supplementary Fig. 1 shows an example of the acceptability division of state-parameter space and the resulting sustainability division.

The division of the parameter space according the safe operating space paradigm is obvious from its definition. Only the cautious policy is a safe policy since it will eventually end up and remain in the prosperous, safe operating space state. The risky policy switches recurrently between the prosperous and the degraded which makes it, by definition, not safe.

**Conversion of timescales to transition probabilities**. Let $p$ be the probability per time step that a system state will transition into another state. The average number of time steps the system will be in that state is given by $\langle N \rangle = (1-p)/p$. Inverting yields $p = 1/(\langle N \rangle + 1)$. We map a model time step to a year. Thus, a collapse time scale of e.g., 50 years corresponds to a collapse probability of $\delta \approx 0.02$. Supplementary Tab. 1 shows the assumed transition timescales and corresponding transition probabilities.

**Code availability**. Python code for the reproduction of the reported results plus interactive versions of the figures can be downloaded from https://github.com/wbarfuss/Paradigms.

**Data availability**. Data sharing not applicable to this article as no datasets were stored on disk during the production of the figures (see Code availability).

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

## Acknowledgements

This work was developed in the context of the COPAN project on Coevolutionary Pathways in the Earth system at the Potsdam Institute for Climate Impact Research. The authors are grateful for financial support from the Heinrich-Böll-Foundation, the Stordalen Foundation (via the Planetary Boundaries Research Network PB.net), the Earth League's EarthDoc program, the Leibniz Association (project DOMINOES) and the Swedish Research Council Formas (Project Grant 2014-589). We thank David Collste, Jobst Heitzig, Antoine Levesque, Finn Müller-Hansen and Maja Schlüter for discussions and comments on the manuscript.

## Author contributions

W.B. designed and analyzed the model with assistance from J.F.D. and S.L. J.F.D and J.K. supervised the project. All authors wrote the manuscript.

## Additional information

**Competing interests:** The authors declare no competing interests.

