## [Peer review file · Nature Communications]

REVIEWERS' COMMENTS:

Reviewer #1 (Remarks to the Author):

There is a large literature that explores aspects of the question of "When optimization is neither sustainable nor safe in human-environment systems" beyond the citations and discussion provided in the manuscript (such as the recent PNAS paper by Carpenter et al). This literature includes work on models with "tipping points", or the capacity for "regime shifts", just to name a few, by Polasky, Maler, A. de Zeeuw, K. Arrow, C. Perrings, W. Brock, S. Carpenter, M. Scheffer, S. Lade, J.D. Matthias etc.

The paper would be strengthened by a more extensive lit review to better place the insights from the model in the existing literature.

Reviewer #2 (Remarks to the Author):

This is an elegant mathematical exercise demonstrating that there are cases in which economic optimization can be neither sustainable nor safe for a 'human-environment' system with a reversible tipping point that yields no rewards in its degraded state.

The title could be more specific in referring to a tipping point (and/or alternative states) because the analysis is specific to that. Equally the analysis is presumably more general in its application than just human-environment systems, but clearly they provide the key framing for the study.

There is much to like about this work. It is very 'clean' in its conceptualisation, mathematical formulation and presentation. The introduction provides a nice mini-review (albeit with some gaps – see below). The results are interesting, intuitive and qualitatively informative. The claim of providing advice to policy makers may be a bit of a stretch: This certainly isn't the first study suggesting that economic optimisation is not sustainable or safe for the climate. Nevertheless the clear and simple mathematical framing of the issues is very welcome.

I would suggest just minor corrections and also have a few suggestions for the authors to consider:

Title: "for" instead of "in"? See remarks above about referring to tipping point (or similar).

Abstract: could be improved to give specific results, particularly for the application to climate. As it is the claim of "offering policy makers guidance" seems vague without stating actual results.

L35: systems (not system's)

L42: ref 24: Malthus as an influence of current sustainability policy?! This is a stretch.

L47-49: there is lots of existing work addressing tipping points within an optimisation framework and the authors need to recognise and cite some of this. One place to start would be Dasgupta and Maler (editors) 'The economics of non-convex ecosystems' (2003) Environment and Resource Economics.

L69-70: but they are present in some economic analyses, for example recent work adding tipping points to conventional cost-benefit analysis models of the climate problem, particularly DICE, e.g. Lontzek et al. (2015) Nature Climate Change, Cai et al. (2016) Nature Climate Change.

L74: "we are not concerned about intra-generational justice in this article" and yet by considering discounting you are concerned about inter-generational justice. My understanding of Dasgupta's critique of the Stern Review is that this is an inconsistent position: If we worry about inter-generational equity then we must simultaneously and consistently be concerned with intra-

generational equity.

L80-81: I am struggling to think of “immediate rewards” of a specific climate policy as these usually have relatively long pay back times (?)

L97 (and throughout): The “discount factor” is somewhat confusing in that 1 is no discounting as I understand – i.e. higher value is not more discounting but rather the opposite. This either needs explaining up front or some alternative terminology should be adopted.

Figures 2 and 3: why isn't the choice of ρ the same for consistency?

L184-185: There are better references for the timescales of triggering a collapse but the values seem reasonable

L185-186: the jump from specific tipping elements to the whole climate system without any explanation is problematic, and ref 50 is an odd choice to justify a long recovery timescale to warming-induced tipping (and it's not relevant to the aforementioned coral reefs).

L201-202: a collapse of all the world's commercial fisheries? This sounds like scaremongering. Surely a sounder argument could be based on historical experience of collapse of specific fisheries, for which the relevant numbers are known.

L207-213: the application to fallow periods of agriculture seems odd. An agricultural example like e.g. the dustbowl would seem in more in-keeping with the preceding examples and the framework.

SI L50-60: there is an exclamation mark on top of an '=' sign (repeatedly) that looks like an error.

Reviewer #3 (Remarks to the Author):

This article presents a formal comparison of three policy “paradigms” – economic optimization, sustainability, and safe operating space (SOS) - using a stylised agent-environment model. To conduct the comparison, a different reward structure is defined for each of the three policy paradigms to influence the agent's decision-making and its effect on the environment over time. The authors vary discount factors and “critical collapse probability” for each policy paradigm and plot overlaps to find that no paradigm guarantees fulfilling the requirements imposed by another paradigm. They also identify the overlap between “sustainable” and “safe operating space” paradigms to be consistent with Kate Raworth's “Safe and Just Operating Space” (SAJOS), and apply the model to compare policies under simulated collapse and recovery conditions of the climate system, fisheries and agriculture.

I am sympathetic to the goal of this article, which reminds me of Donella Meadows' observation that the ability to transcend paradigms - to use the paradigm that best fits the issue at hand - may be the most powerful leverage point for creating systemic change. The interdisciplinary findings are original and likely to be of interest to the broad sustainability and climate science & policy communities.

The justification for why the research is important elaborated on Lines 60-70 is not particularly convincing, however. The reader would benefit from a clear research question, perhaps before jumping into paradigm background in paragraph 2.

Line 29: I would argue that optimization approaches have emerged from neoclassical economic theory, which makes several simplifying assumptions compared to utilitarianism as an ethical theory.

Line 74: Why are the authors not concerned about intragenerational justice in this article?

The results are fairly dense in formal notation, which makes it difficult for the reader to capture the main original messages. My preference is to relegate as much formal notation as possible to methods/SI text.

A paragraph summarising the important parameters (e.g. what they are, why they are important, and how they vary) using words would be helpful in the description of the model, i.e. lines 77-97. Are the results particularly sensitive to individual parameters? A discussion of model sensitivity to the chosen parameters is lacking here or in the methods/SI.

A point that should be acknowledged in my view is that the definition used by the authors for the sustainability paradigm in equation (2) is based upon the notion of "weak sustainability" whereby the use of a single discounted value reduces natural, manufactured and other forms of capital to substitutes rather than complements. Advocates of "strong sustainability" generally reject this approach and argue that natural capital should be accounted for separately from other forms of capital.

Figure 2 is interesting but it is not clear to me why the other parameters are set to $p=0.1$ and $r_l / r_h = 0.5$, and $r_{min} / r_h = 0.2$.

Line 119: In practice, the safe operating space as defined by Rockstrom and colleagues is not "bounded by tipping thresholds" because nobody can say where those thresholds actually lie ex ante. The planetary boundaries framework is quite clear that the SOS is based on the precautionary principle whereby each boundary is placed at the lower end of an uncertainty range. This choice is explicitly recognised to be normative and is therefore somewhere in between equation (2) and equation (3) (i.e. there is a minimum acceptable "value", but the units are not directly comparable to the net present value of the agent's rewards).

Line 128: This is an example of how the formal notation in parentheses confuses more than it illuminates. If the results depend so much upon the model parameters, it would be better to walk the reader through them using words. In particular, I didn't see descriptions of the ratios r_l/r_h and r_{min}/r_h anywhere besides numerically in figure captions.

I quite like Figure 3a in terms of the stylized vision of the paradigm overlap but I would also be interested to see how the estimated results for each policy are clustered after running the model. Would it be possible to visualise the overlap in the model output for each policy numerically using principal component/cluster analysis? I would expect ellipses rather than perfect circles, at the very least. It would be particularly interesting to show the size and shape of the "safe and just operating space", if it exists.

Lines 155-170: As above, it is strange to see that a policy can be safe but not sustainable and vice versa to someone who does not hold the weak sustainability viewpoint.

The discussion section presents new and interesting results from the model using collapse/recovery probabilities that arguably apply to the climate system, fisheries and agriculture. To me, these new results could be a sub-section of the Results section and leave the Discussion for the final few paragraphs and possibly more detail on the broader implications. What do the authors think their results mean for the widespread use of optimizing Integrated Assessment Models to inform climate policy? What about optimal Maximum Sustainable Yields in fisheries management?

Response to referees

Tracking number: NCOMMS-18-03512

Reviewer #1:

There is a large literature that explores aspects of the question of "When optimization is neither sustainable nor safe in human-environment systems" beyond the citations and discussion provided in the manuscript (such as the recent PNAS paper by Carpenter et al). This literature includes *work on models with "tipping points", or the capacity for "regime shifts"*, just to name a few, by Polasky, Maler, A. de Zeeuw, K. Arrow, C. Perrings, W. Brock, S. Carpenter, M. Scheffer, S. Lade, J.D. Matthias etc.

The paper would be strengthened by a more extensive lit review to better place the insights from the model in the existing literature.

⇒ We thank the reviewer for the time and effort of reviewing our manuscript. Thank you for your remark to strengthen our paper by referencing more to existing literature on the topic of management and regime shifts/tipping elements. With the revised version of the ms we framed our work more appropriately by including further references to this literature. For example, we additionally cited

- **Stephen Polasky, Stephen R Carpenter, Carl Folke, and Bonnie Keeler. Decision-making under great uncertainty: environmental management in an era of global change. Trends in ecology & evolution, 26(8):398–404, 2011.**
- **Anne-Sophie Crépin, Reinette Biggs, Stephen Polasky, Max Troell, and Aart de Zeeuw. Regime shifts and management. Ecological Economics, 84:15–22, 2012.**
- **Stephen Polasky, Aart De Zeeuw, and Florian Wagener. Optimal management with potential regime shifts. Journal of Environmental Economics and management, 62(2):229–240, 2011.**
- **Karl-Göran Mäler and Chuan-Zhong Li. Measuring sustainability under regime shift uncertainty: a resilience pricing approach. Environment and Development Economics, 15(6):707–719, 2010.**
- **Partha Dasgupta and Karl-Göran Mäler. The economics of non-convex ecosystems, volume 4. Springer Science & Business Media, 2006.**

Reviewer #2:

This is an elegant mathematical exercise demonstrating that there are cases in which economic optimization can be neither sustainable nor safe for a 'human-environment' system with a reversible tipping point that yields no rewards in its degraded state.

⇒ We thank the reviewer for her or his assessment and the quality of the comments provided. We are confident that they greatly improved the quality of our ms.

The title could be more specific in referring to a tipping point (and/or alternative states) because the analysis is specific to that. Equally the analysis is presumably more general in its application than just human-environment systems, but clearly they provide the key framing for the study.

⇒ You are absolutely right. We adjusted the title to "When optimisation in human-environment tipping elements is neither sustainable nor safe" in the revised version of our ms to be more specific to our work

There is much to like about this work. It is very 'clean' in its conceptualisation, mathematical formulation and presentation. The introduction provides a nice mini-review (albeit with some gaps – see below). The results are interesting, intuitive and qualitatively informative. The claim of providing advice to policy makers may be a bit of a stretch: This certainly isn't the first study suggesting that economic optimisation is not sustainable or safe for the climate. Nevertheless the clear and simple mathematical framing of the issues is very welcome.

I would suggest just minor corrections and also have a few suggestions for the authors to consider:

- Title: "for" instead of "in"? See remarks above about referring to tipping point (or similar).
⇒ **We reformulated the title in the revised version of the ms.**
- Abstract: could be improved to give specific results, particularly for the application to climate. As it is the claim of "offering policy makers guidance" seems vague without stating actual results.
⇒ **We shortened the abstract (also to fit editorial constraints) to describe our work more accurately and to highlight the result that the climate system might reside in a parameter regime where economic optimization is neither sustainable nor safe.**
- L35: systems (not system's)
⇒ **Corrected.**
- L42: ref 24: Malthus as an influence of current sustainability policy?! This is a stretch.
⇒ **Originally this was meant as reference to the early beginnings of what may today be called sustainability discourse. However, we agree with your assessment and decided to remove the reference.**
- L47-49: there is lots of existing work addressing tipping points within an optimisation framework and the authors need to recognise and cite some of this. One place to start would be Dasgupta and Maler (editors) 'The economics of non-convex ecosystems' (2003) Environment and Resource Economics.
⇒ **We thank the reviewer for pointing out this material, which we acknowledged in our revised version of the ms (L. 63 in revised ms with changes)**
- L69-70: but they are present in some economic analyses, for example recent work adding tipping points to conventional cost-benefit analysis models of the climate problem, particularly DICE, e.g. Lontzek et al. (2015) Nature Climate Change, Cai et al. (2016) Nature Climate Change.
⇒ **Again, we are thankful for these references, which we referenced to in the revised version of our manuscript (L. 63 in revised ms with changes). We decided further to remove the line, that was L.69-70 in the original submission, since it is not essential at this position. Content-wise it is already covered at original L.47-49.**
- L74: "we are not concerned about intra-generational justice in this article" and yet by considering discounting you are concerned about inter-generational justice. My understanding of Dasgupta's critique of the Stern Review is that this is an inconsistent position: If we worry about inter-generational equity then we must simultaneously and consistently be concerned with intra-generational equity.
⇒ **In this article we focus on one decision making agent in contrast to a multiaгент setting. This is why intra-generational matters are not the topical focus of this article. We clarified our unfavorable sentence in the revised version of the ms to be more understandable (L. 88 in revised ms with changes). Thank you for pointing this out.**
- L80-81: I am struggling to think of "immediate rewards" of a specific climate policy as these usually have relatively long pay back times (?)
⇒ **The immediate reward of a climate friendly policy could be comparable less economic output compared to a climate un-friendly policy. This is what we meant**

by this statement. We updated the example in the revised version of the ms to be more understandable. (L. 114 in revised ms with changes)

- L97 (and throughout): The “discount factor” is somewhat confusing in that 1 is no discounting as I understand – i.e. higher value is not more discounting but rather the opposite. This either needs explaining up front or some alternative terminology should be adopted.
⇒ **Thank you for pointing out this reading of the discount factor. Since the discount factor is an established term (in e.g. Markov Decision Processes) we added an explanation right after introducing it (L. 123 in revised ms with changes)**
- Figures 2 and 3: why isn't the choice of rho the same for consistency?
⇒ **Thank you for this important hint. We updated the figures with consistent parameter choices.**
- L184-185: There are better references for the timescales of triggering a collapse but the values seem reasonable
⇒ **We additionally referenced to the tipping paper by Lenton et al. in which several timescales of relevant tipping elements are listed (L. 262 in revised ms with changes)**
- L185-186: the jump from specific tipping elements to the whole climate system without any explanation is problematic, and ref 50 is an odd choice to justify a long recovery timescale to warming-induced tipping (and it's not relevant to the aforementioned coral reefs).
⇒ **Thank you for this important point to consider. We explained our approach and stated its limitations more clearly. With ref 50 we wanted to show that human perturbations have already altered the trajectory of the Earth system in a fundamental way. We added an explanation of our reasoning and an additional reference in the revised version of the ms (Paragraph from L. 268 on in revised ms with changes)**
- L201-202: a collapse of all the world's commercial fisheries? This sounds like scaremongering. Surely a sounder argument could be based on historical experience of collapse of specific fisheries, for which the relevant numbers are known.
⇒ **We agree that our formulation was unfortunate at this place. We reformulated this paragraph in the revised version of the ms. There, we added additional references to the case of the Baltic Sea that support our choice of timescales (Paragraph from L. 283 on in revised ms with changes). Additionally we added the idea of a more detailed link between empirical case and our stylized model to the discussion in the revised version of the ms (Paragraph from L. 360 on in revised ms with changes)**
- L207-213: the application to fallow periods of agriculture seems odd. An agricultural example like e.g. the dustbowl would seem in more in-keeping with the preceding examples and the framework.
⇒ **The example of a dustbowl is an excellent one to consider and indeed reflecting upon the agriculture example we decided to adapt the example to “land degradation by farming” which fit much better to our model. (Paragraph from L. 302 on in revised ms with changes). Thank you very much for this suggestion.**
- SI L50-60: there is an exclamation mark on top of an ‘=’ sign (repeatedly) that looks like an error.
⇒ **The exclamation mark above the equal sign shall denote “should be equal to”. However, to avoid confusion we changed the “!” to “set” over the “=” sign which will hopefully be more descriptive. Thank you for this remark.**

Reviewer #3:

This article presents a formal comparison of three policy “paradigms” – economic optimization, sustainability, and safe operating space (SOS) - using a stylised agent-environment model. To conduct the comparison, a different reward structure is defined for each of the three policy paradigms to influence the agent’s decision-making and its effect on the environment over time. The authors vary discount factors and “critical collapse probability” for each policy paradigm and plot overlaps to find that no paradigm guarantees fulfilling the requirements imposed by another paradigm. They also identify the overlap between “sustainable” and “safe operating space” paradigms to be consistent with Kate Raworth’s “Safe and Just Operating Space” (SAJOS), and apply the model to compare policies under simulated collapse and recovery conditions of the climate system, fisheries and agriculture.

I am sympathetic to the goal of this article, which reminds me of Donella Meadows’ observation that the ability to transcend paradigms - to use the paradigm that best fits the issue at hand - may be the most powerful leverage point for creating systemic change. The interdisciplinary findings are original and likely to be of interest to the broad sustainability and climate science & policy communities.

⇒ We are very thankful to the reviewer for the time and effort she or he put into reviewing our work. From our point of view our ms has immensely benefited by the comments and suggestions provided.

- The justification for why the research is important elaborated on Lines 60-70 is not particularly convincing, however. The reader would benefit from a clear research question, perhaps before jumping into paradigm background in paragraph 2.
⇒ Thank you for this important point. As suggested we added a research question to the introduction in the revised version of our ms (Paragraph from L. 36 on in revised ms with changes).
- Line 29: I would argue that optimization approaches have emerged from neoclassical economic theory, which makes several simplifying assumptions compared to utilitarianism as an ethical theory.
⇒ As with Malthus (see above) referencing to utilitarianism was meant to cover the origins of the concept of utility. However, as with Malthus we decided to remove these reference in order to avoid confusion.
- Line 74: Why are the authors not concerned about intragenerational justice in this article?
⇒ In this article we focus on one decision making agent in contrast to a multigent setting. This is why intra-generational matters are not the topical focus of this article. We clarified our unfavorable sentence in the revised version of the ms to be more understandable (L. 88 in revised ms with changes).
- The results are fairly dense in formal notation, which makes it difficult for the reader to capture the main original messages. My preference is to relegate as much formal notation as possible to methods/SI text.
⇒ We tried to make our work more accessible with the revised version of the ms, for example by adding a paragraph summarizing the model parameters as you suggested. We also added an explanation of the discount factor and discussed the definition Safe Operating Space more in detail and removed original Eq. 3.
- A paragraph summarising the important parameters (e.g. what they are, why they are important, and how they vary) using words would be helpful in the description of the model, i.e. lines 77-97. Are the results particularly sensitive to individual parameters? A discussion of model sensitivity to the chosen parameters is lacking here or in the methods/SI.

⇒ Thank you for this excellent suggestion. We added such a paragraph (L. 163 in revised ms with changes)

A model sensitivity analysis is given in this paper by plotting two dimensional subspaces of the parameter space, showing how sensitive the model is by varying the two parameters shown. The analytical expressions in the SI (Methods section in the revised version of the ms) give the full dependencies of our policy paradigms classification on the parameters. Additionally, we updated the last figure of the ms, where now color gradients between the paradigm combinations regimes indicate the probability to be in a certain regime under parameter uncertainty.

- A point that should be acknowledged in my view is that the definition used by the authors for the sustainability paradigm in equation (2) is based upon the notion of “weak sustainability” whereby the use of a single discounted value reduces natural, manufactured and other forms of capital to substitutes rather than complements. Advocates of “strong sustainability” generally reject this approach and argue that natural capital should be accounted for separately from other forms of capital.

⇒ Thank you very much for this important point which we included after our definition of sustainability in the revised version of our ms (L. 148 in revised ms with changes). From our understanding, a “strong sustainability” paradigm, translated to our model system, would be exactly the combination of the safe and sustainable paradigm. So, our definition of sustainability shall not be understood as our personal preference of “weak” over “strong” sustainability, but as conceptual division.

- Figure 2 is interesting but it is not clear to me why the other parameters are set to $p=0.1$ and $r_l / r_h = 0.5$, and $r_{min} / r_h = 0.2$.

⇒ Figure 2 shall give an example of the model’s parameter space. The remaining parameters were chosen for illustrational purposes. We updated the Figure caption in the revised version of the ms with this explanation.

- Line 119: In practice, the safe operating space as defined by Rockstrom and colleagues is not “bounded by tipping thresholds” because nobody can say where those thresholds actually lie ex ante. The planetary boundaries framework is quite clear that the SOS is based on the precautionary principle whereby each boundary is placed at the lower end of an uncertainty range. This choice is explicitly recognised to be normative and is therefore somewhere in between equation (2) and equation (3) (i.e. there is a minimum acceptable “value”, but the units are not directly comparable to the net present value of the agent’s rewards).

⇒ Thank you very much for this excellent point. We believe this opens promising paths for future work considering explicitly the level of normative risk tolerance versus the level of how much worse must a tipped “degraded” state be to be considered as unfavorable or “disastrous”. For this article we added a paragraph explaining the more detailed SOS definition and positioning our choice of no risk tolerance at all within this line of argument (Paragraph from L. 150 on in revised ms with changes).

- Line 128: This is an example of how the formal notation in parentheses confuses more than it illuminates. If the results depend so much upon the model parameters, it would be better to walk the reader through them using words. In particular, I didn’t see descriptions of the ratios r_l/r_h and r_{min}/r_h anywhere besides numerically in figure captions.

⇒ Thank you for pointing this out. We added a paragraph summarizing all parameters of the model in the revised version of the ms. By doing so we hope the formal notation in this line becomes more clear.

- I quite like Figure 3a in terms of the stylized vision of the paradigm overlap but I would also be interested to see how the estimated results for each policy are clustered after running the model.

Would it be possible to visualise the overlap in the model output for each policy numerically using principal component/cluster analysis? I would expect ellipses rather than perfect circles, at the very least. It would be particularly interesting to show the size and shape of the “safe and just operating space”, if it exists.

⇒ Thank you for this excellent suggestion. In terms of size, we added an additional plot in the revised version of the ms showing the relative volumes of all eight paradigm combinations in the five-dimensional space of all possible model parameter settings.. This should correspond to a model based version of the paradigm overlap.

In terms of the shapes of the paradigm combinations (including the shape of the SAJOS) we decided against a dimension reduction of the five-dimensional parameter space. We did so because the principal components consisting of combinations of our already conceptual parameters would add an additional level of conceptualisation we feel is not appropriate for the scope of this paper.

However, it might be a promising path for future work.

- Lines 155-170: As above, it is strange to see that a policy can be safe but not sustainable and vice versa to someone who does not hold the weak sustainability viewpoint.
⇒ We discussed the similarity of our sustainability definition with the notion of weak sustainability as the reason for why these regimes do not cluster completely. As we wrote in the revised version of the ms, our reasoning is not to favor a weak over a strong sustainability paradigm. But a weak sustainability paradigms is somewhat within the lines of our stylized model which also knows only one form of reward (From L. 220 on in revised ms with changes).
- The discussion section presents new and interesting results from the model using collapse/recovery probabilities that arguably apply to the climate system, fisheries and agriculture. To me, these new results could be a sub-section of the Results section and leave the Discussion for the final few paragraphs and possibly more detail on the broader implications. What do the authors think their results mean for the widespread use of optimizing Integrated Assessment Models to inform climate policy? What about optimal Maximum Sustainable Yields in fisheries management?
⇒ Thank you very much for this excellent suggestion. We put the real-world examples as a sub-section to the results in the revised version of the ms and elaborated on the implications of our work in the discussion section.